# Cyberbullying and Psychopathological Behaviors in Spanish Secondary Education Students

**DOI:** 10.3390/healthcare11243162

**Published:** 2023-12-13

**Authors:** Ángel Enrique Contreras-Piqueras, Cecilia Ruiz-Esteban, Inmaculada Méndez

**Affiliations:** Department of Evolutionary Developmental and Educational Psychology, University of Murcia, 30100 Murcia, Spain; angelenrique.contrerasp@um.es (Á.E.C.-P.); cruiz@um.es (C.R.-E.)

**Keywords:** cyberbullying, mental health, psychopathologies, prevention, adolescents

## Abstract

Cyberbullying is considered a serious health problem that mainly affect adolescents, with different characteristics depending on the role in which they are directly involved. The objective of our research was to analyze the sociodemographic characteristics and psychopathological behaviors among the roles of those involved in cyberbullying (victims, aggressors, bystanders and aggressive victims). The study participants were 280 students aged between 12 and 17 years (61.8% female) from the Region of Murcia, Spain. The Cyberbullying Screening of Bullying among Peers, the Clinical Analysis Questionnaire and a sociodemographic survey were used. Hierarchical regression analysis was used. This study showed that the roles of those directly involved in cyberbullying correlate with psychopathological behaviors. It is a priority to promote prevention programs aimed at improving cyberbullying among students.

## 1. Introduction

Since the emergence of new technological advances, especially the Internet and social and communicative media, along with the ease of access to information due to the rise of mobile phones (also known as smartphones), our understanding of the world has changed without us realizing it, and the way we relate to the environment and people has also experienced a new conception [1]. Currently, technologies are an important part of people’s daily lives [2,3]. They are agents of socialization, instruments of communication and a new way of enjoying leisure time, which is the positive part of technological tools. However, there is a negative side generated by their inappropriate use, since they are sometimes used to commit aggressions, crimes and harassment by aggressors, thus creating a new version of harassment towards victims, which can occur in different contexts in which people develop, we are not referring to cyberbullying [4,5,6]. We can define cyberbullying as an aggressive act of an intentional nature, carried out through the use of new technologies [4,7]. Among the main characteristics of cyberbullying, the following can be highlighted [8]:Deliberation: the aggressor must assume that he/she has hurt the victim, and this leads to follow-up.Redundancy: in cyberbullying, as in bullying, the hostility must be repeated at least once or twice. We can capture redundancy in cyberbullying by assuming that the hostility is seen a few times by others or by the involved individuals themselves.Control of violence: the individual cannot do anything against the aggression, the person in question does not have the power to delete an image, a video or a recently sent message from the network.Absence of physical and social contact between the individuals: there is no physical contact between the aggressor and his victim, so that the latter’s reaction cannot be predicted; however, it obviously causes him discomfort through problematic, uninhibited, violent and reckless behavior.Open channel: aggression can occur at any time, 24 h a day, 7 days a week.

It has been questioned whether cyberbullying and traditional bullying are two isolated situations or whether they are one and the same phenomenon that is carried out using a different medium [7,9]. Regarding this point, Íñiguez et al. [10] express that traditional bullying continues with cyberbullying, hence the importance of examining the predictors of this behavior. Specifically, in the relationships that take place in the school space, these behaviors of cyberbullying are frequent [11,12,13]. Data on cyberbullying derived from reports prepared by Save the Children [14,15] in Spain indicate that the figures continue to be worrying both for schoolchildren involved in recent months as aggressors (3.3% in cyberbullying) and for those in the role of victim (6.9% in cyberbullying).

Cyberbullying is a social phenomenon that presents diverse manifestations, the aim of which is to cause physical and psychological harm to other people [16]. One of the most difficult issues of cyberbullying is its cross-cutting nature; for example, it influences the entire adolescent’s climate: peers, studies, family, social relationships, character, etc. [5,8,17].

There are various forms of cyberbullying, including:Mental harassment, with which the aim is to make a generalized, constant and vindictive exclusion of the victim in order to embarrass him/her.Search for private data of the victim, to use them to assault the person in question.Grooming, in which an adult assumes forms of behavior to gain the trust of a minor and physically tempt him or her.Sextortion or blackmail involves coercing a person by threatening to reveal compromising photos, recordings, or materials in their possession.Message harassment includes various forms of communication such as calls, messages, or correspondence aimed at affronting, attacking, or tormenting the targeted individual.

Another form of virtual interaction is cyberchisme, which involves supposedly harmless or innocent dialogue but includes evaluative judgments about a known absent person [18].

Cyberbullying is a significant health issue, particularly among adolescents, and has distinct characteristics depending on the role they are directly involved in [19]. As interpersonal interactions occur, openness can become a persistent form of harassment [17,20]. Victims often exhibit depressive symptoms, low emotional intelligence, fear, social isolation, shyness, and introversion [21,22,23,24,25]. Aggressors typically display antisocial and criminal behaviors, a lack of empathy, extraversion, a lack of self-control, and cruelty [7,23,26]. Prolonged bystander roles are often associated with internalization of antisocial behaviors, loss of empathy, guilt, and desensitization [7,23,27].

Similarly, cyberbullying has a multidimensional character related to socio-cultural, moral, affective, and motivational factors in both aggressors and victims [28]. Among the influential factors in cyberbullying is social status; aggressive behavior is often found in individuals belonging to the medium to high socioeconomic status, which may be attributed to their greater access to technological tools [29].

Therefore, this study aims to objectively analyze the socio-demographic characteristics and psychopathological behaviors among various roles involved in cyberbullying, including victims, aggressors, bystanders, and aggressive victims.

The specific objective is to investigate the:Predictive capacity of the sociodemographic traits and psychopathological behaviors in the victim’s cyberbullying profile (Aim 1).Predictive capability of the sociodemographic features and psychopathological behaviors in the aggressor’s profile in cyberbullying (Aim 2).Predictive capability of the sociodemographic features and psychopathological behaviors in the bystander’s profile in cyberbullying (Aim 3).Predictive capacity of the sociodemographic characteristics and psychopathological behaviors in the profile of aggressive victimization in cyberbullying (Aim 4).

## 2. Materials and Methods

### 2.1. Participants

This study involved 280 students, ranging in age from 12 to 17 years, from three educational centers in the Murcia region of Spain. Additionally, 13.2% of the students had repeated a grade, which resulted in some students being over the age of 16. Of these students, 61.8% were male and 38.2% were female. Regarding the family’s socioeconomic and cultural level, 81.1% were classified as having a medium level, 16.4% had a high level, and 2.5% had a low level. In total, 33.6% of the parents or legal guardians work in the tertiary sector providing various services, 26.1% work in the primary sector, and 18.6% in the secondary sector. Furthermore, 21.8% of respondents indicated they were not certain of their parents’ and/or guardians’ work activity location. In terms of participants’ living arrangements, 74.3% reside with their parents and siblings, 11.4% with only their parents, 8.9% with parents, siblings, and other relatives (such as grandparents, aunts, and uncles), 5% with only their mother, and 0.4% with only their father. A total of 99.6% of the participants had access to the internet at home, whereas only 0.4% did not have access.

### 2.2. Instruments

Several instruments were used for this study and are explained below.

First, a sociodemographic scale, developed ad hoc, was used to measure the following variables: age (12–13 years, 14–15 years, 16 to 17 years), gender (male or female), educational level (first grade, second grade, third grade, and fourth grade), grade repetition (repeating or not repeating a school year), perceived socio-economic and cultural level of the family (high, medium, or low), work activity of the parents (primary, secondary, and tertiary sector), family cohabitation (persons with whom they live or share the house: both parents, only their father, only their mother, with parents and siblings, with parents, siblings, and another relative (uncles, grandparents)), have internet access at home (yes or no).

To study variables related to cyberbullying, we used the “Cyberbullying. Screening of Peer Bullying” questionnaire by Garaigordobil [30], which has been validated for the Spanish population. This tool enables the assessment of traditional bullying and cyberbullying behaviors. For this investigation, only the section on cyberbullying was utilized. This section comprises 15 items that reference potential cyberbullying conduct within a group. It enables the acquisition of information concerning the roles that participate in cyberbullying: victimization, aggression, observation, and aggressive victimization. As an example of cyberbullying items, when acting as a victim, “Have offensive or insulting messages been sent to you via cell phone or internet?”, as an aggressor, “Have offensive or insulting messages been sent via cell phone or internet by you?” and as bystander “Have you witnessed offensive and insulting messages being sent via cell phone or internet?”. Aggressive victimization is determined by adding the situations of cyberbullying as a victim and cyberbullying as an aggressor. Cronbach’s alpha reliability values for the overall scale are adequate high (α = 0.91), and the values for each subscale are as follows: cyberbullying as a victim (α = 0.82), cyberbullying as an aggressor (α = 0.91), and cyberbullying as bystander (α = 0.87) [31]. In our study, the Cronbach’s alpha reliability of the Cyberbullying Test was α = 0.88 for the total test, and the values for each subscale were as follows: cyberbullying as victim (α = 0.83), cyberbullying as aggressor (α = 0.92), and cyberbullying as bystander (α = 0.86).

Finally, Krug’s [32] Clinical Analysis Questionnaire (CAQ), validated in the Spanish population, was used. It is an instrument that allows the evaluation of psychopathological personality traits or behaviors. The instrument allows the assessment of 12 scales through its 144 items: Hypochondriasis (degree of preoccupation with health and disturbances such as aches or pains), Suicidal depression (degree of feelings of dissatisfaction and disgust with life), Agitation (degree of excitement), Anxious depression (level of tension and restlessness), Low-energy depression (level of energy or vitality), Guilt–resentment (level of criticism of self), Apathy–withdrawal (interpersonal relationship skills), Paranoia (level of suspiciousness or suspicion), Psychopathic deviance (level of sensation seeking and strong emotions), Schizophrenia (degree of behavioral deviations involving a severe maladjustment of cognitive level and mood and loss of contact with reality), Psychasthenia (degree of obsessive behaviors or fears), and Psychological maladjustment (degree of self-confidence of the person). Example of items: “When I get discouraged, I find it hard to recover”. The instrument has adequate reliability values on the original scale average values of α = 0.80 for the total scale, with the following values for each of the scales: Hypochondriasis (α = 0.85), Suicidal depression (α = 0.74), Agitation (α = 0. 85), Anxious depression (α = 0.73), Low-energy depression (α = 0.75), Guilt–resentment (α = 0.71), Apathy–withdrawal (α = 0.67), Paranoia (α = 0.86), Psychopathic deviance (α=.84), Schizophrenia (α = 0.90), Psychasthenia (α = 0.75), and Psychological maladjustment (α = 0.84) [32]. In our study, it was α = 0.95 for the total scale, with the following values for each of the scales: Hypochondriasis (α = 0.84), Suicidal depression (α = 0.87), Agitation (α = 0.67), Anxious depression (α = 0.82), Low-energy depression (α = 0. 81), Guilt–resentment (α = 0.72), Apathy–withdrawal (α = 0.71), Paranoia (α = 0.87), Psychopathic deviance (α= 0.85), Schizophrenia (α = 0.88), Psychasthenia (α = 0.83), and Psychological maladjustment (α = 0.88).

### 2.3. Procedure

The following sections describe the research phases in detail.

Authorization from the Research Ethics Committee at the University of Murcia (ID: 3764/2022) was requested before data collection commenced. Upon receiving authorization, three participating centers were conveniently selected. A preliminary meeting was conducted with the management teams to outline the research objectives and administration process for the instruments. First, consent was obtained from parents and minors. The evaluation instruments were completed during class sessions in the classrooms.

Afterwards, high school tutors and guidance counselors were asked for collaboration in granting access to students and administering the evaluations. Participants who did not have family consent or informed assent from minors were excluded from the research. Additionally, poorly or partially completed instruments (around 15–20 per education center) were also excluded.

The instruments were completed in four 60 min sessions, divided into four groups across each educational center with approximately 28–30 students in each classroom. Anonymity, confidentiality, and voluntary participation were ensured throughout the process. Tutors and/or guidance counselors administered the tests at various educational centers in accordance with the directions given by the principal investigator.

Data were collected during the 2021/2022 academic year, and relevant data analysis was subsequently conducted. Any instruments that were wrongly or incompletely completed were excluded. Additionally, data were absent for students who did not attend class on the day the instruments were completed.

The research design was descriptive, quantitative, transactional, and non-experimental. 

### 2.4. Data Analysis

First of all, it was necessary to detect missing cases in the database in order to carry out an adequate treatment of them. Secondly, descriptive techniques such as frequencies, percentages, means, and standard deviations were used. Thirdly, to obtain the relationship between the cyberbullying variables and psychopathological behaviors, Pearson’s bivariate correlation test was carried out. Fourth, in order to meet the objectives set out, a stepwise hierarchical regression analysis was performed using the method introduced to contrast the predictive power of the independent variables of the study (sociodemographic variables such as gender and age) with respect to the dependent variable under study (victim in cyberbullying, aggressor in cyberbullying, bystander in cyberbullying, and aggressor victim in cyberbullying). The analyses were performed in the IBM (Armonk, NY, USA) program SPSS v.28.

## 3. Results

Table 1 presents the descriptive values calculated to understand the distribution of adolescents in the analyzed variables. The subscale values of the CAQ instrument range from a minimum score of 0 to a maximum of 24. The highest values in the cyberbullying test subscales are found in aggressive victimization, which encompasses both cyberbullying as a victim and as an aggressor. Following this is cyberbullying as a bystander, then cyberbullying as a victim and cyberbullying as an aggressor.

Table 2 shows the Pearson correlations that were statistically significant between the roles of those directly involved in cyberbullying and psychopathological behaviors. In the case of cyberbullying as a victim, significant psychopathological traits or behaviors stand out with all of them except agitation. In the case of cyberbullying as an aggressor, all psychopathological behaviors stand out as significant with the exception of Agitation, Anxious depression, Guilt–resentment, and Psychopathic deviance. In the role of cyberbullying as aggressive victimization, all behaviors or psychopathological traits stand out as significant with the exception of Agitation and Psychopathic deviance. In the role of cyberbullying as bystander, all the psychopathological behaviors stand out as significant, with the exception of Anxious depression.

For the first objective (“To determine the predictive capacity of sociodemographic characteristics and psychopathological behaviors in the profile of the victim in cyberbullying”), a hierarchical regression analysis was conducted to predict the characteristics of victims in cyberbullying by examining the predictive capacity of sociodemographic characteristics and psychopathological behaviors. This study controlled for gender and age, which explained only 1.1% of the variance.

Thus, using the cyberbullying victim profile as a criterion, we included variables that provided information on psychopathological behaviors as a second step, which accounted for 28.5% of the variance, as demonstrated in Table 3. These results provide valuable insights for detecting and preventing cyberbullying. It is noteworthy that the standardized Beta coefficients indicated the noteworthy variables to be Low-energy depression (Beta = 0.186; t = 1.988; *p* = 0.048) and Paranoia (Beta = 0.248; t = 3.347; *p* = 0.001). The results showed significant relationships between victimization and Psychopathic deviance (Beta = 0.149; t = 2.446; *p* = 0.015), Schizophrenia (Beta = 0.267; t = 3.065; *p* = 0.002), and Psychasthenia (Beta = 0.148; t = 2.811; *p* = 0.005).

A hierarchical regression analysis was conducted to predict the characteristics of cyberbullying aggressors, in accordance with Aim 2, which aimed to determine the predictive capacity of sociodemographic traits and psychopathological behaviors. The sociodemographic variables of this study, such as gender and age, were utilized for this purpose, explaining a mere 0.4% of the variance.

Thus, using the cyberbullying aggressor profile as a criterion, the second step comprised psychopathological behaviors, which accounted for 9.1% of the variance (refer to Table 4). It is worth noting that the standardized Beta coefficients showed that the significant variables were Schizophrenia (Beta = 0.236; t = 2.403; *p* = 0.017) and Psychasthenia (Beta = 0.122; t = 2.066; *p* = 0.040).

A hierarchical regression analysis was conducted to predict the characteristics of schoolchildren who assume the role of bystander in cyberbullying, in accordance with Aim 3, which aimed to determine the predictive ability of socio-demographic and psychopathological behaviors in the bystander’s profile. In this regard, this study’s sociodemographic variables (gender and age), which account for 1% of the variance, were held constant.

Based on the cyberbullying bystander profile, the second step incorporated psychopathological behaviors, which accounted for 14.6% of the variance (see Table 5). In this regard, it should be noted that the standardized Beta coefficients showed that the significant variables were Schizophrenia (Beta = 0.197; t = 2.067; *p* = 0.039) and Psychasthenia (Beta = 0.173; t = 3.015; *p* = 0.003).

Model R², adjusted r-squared, and R² change were analyzed using a hierarchical regression analysis to predict the characteristics of aggressive victims in cyberbullying in relation to the fourth aim, which was to determine the predictive ability of sociodemographic characteristics and psychopathological behaviors. This was achieved by controlling for the sociodemographic variables of this study, namely gender and age, which explained 0.5% of the variance.

Subsequently, utilizing the aggressive cyberbullying victim profile as a criterion, the second step involved examining psychopathological behaviors that account for 19.6% of the variance, as displayed in Table 6. The standardized beta coefficients reveal that the variables indicating significance were Paranoia (Beta = 0.154; t = 1.967; *p* = 0.050), Schizophrenia (Beta = 0.271; t = 2.936; *p* = 0.004) and Psychasthenia (Beta = 0.146; t = 2.622; *p* = 0.009).

## 4. Discussion

This study demonstrates a correlation between the roles of cyberbullying participants and psychopathological behaviors [15,20].

Participants who assumed the victim role displayed psychopathological behaviors such as depressive symptomatology, loneliness leading to energy loss, paranoia accompanied by neurosis and fear, psychopathic deviance causing inhibition, schizophrenia leading to low interpersonal relationships and neurosis, and psychasthenia, characterized by anxiety and neuroticism [21,22,23,24]. These results are in line with previous research that shows that the role of the victim in cyberbullying shows personality traits or psychopathological behaviors characterized by depression, anxiety, low interpersonal skills, loneliness, etc. The profile of the aggressor exhibited psychopathological behaviors such as schizophrenia (difficulties in managing impulses and social relationships) and psychasthenia (anxiety and obsessive behaviors). These findings align with prior research indicating that aggressors typically display traits or behaviors that include a lack of impulse control, anxiety, impulsivity, irritability, externalizing problems, and other issues that may be linked to aggression [23,24,26]. Regarding the role of aggressive victims in cyberbullying, they exhibited paranoia (suspiciousness, neurosis, and feelings of injustice), schizophrenia (feelings of unreality and distancing of interpersonal relationships), and psychasthenia (anxiety and neuroticism). These findings align with previous research illustrating that victims of cyberbullying display personality traits or psychopathological behaviors, such as anxiety, suspicion, low interpersonal skills, and loneliness [17,23,24,25]. In the bystander’s profile, schizophrenia (which involves interpersonal relationships where estrangement prevails) and psychasthenia (which involves low self-control and anxiety) were identified as psychopathological behaviors. These results are consistent with previous findings indicating that observers tend to exhibit personality traits or behaviors with low anxiety and a lack of solidarity, alongside moral disconnection and feelings of submission and fear [23,24,27]. Although sociodemographic variables such as gender and age were controlled for in our study, they did not show any significance in the different roles of individuals involved in cyberbullying. This suggests that the relationship between psychopathological behaviors and cyberbullying is independent of both age and gender, with both boys and girls participating in various cyberbullying roles [20,29].

Therefore, the present study provides information that supports the results obtained by Alonso and Romero [19] who demonstrated that cyberbullying has different characteristics depending on the roles of the parties directly involved. The psychopathological behaviors associated with cyberbullying differ according to the profile, which could aid in developing prevention and intervention programs. Thus, promoting programs that improve cybercoexistence among schoolchildren is essential, including emotional management, interpersonal relationships, and social skills. Additionally, it is important to involve bystanders who are often passive in these situations. Therefore, cyberbullying prevention workshops in schools are necessary to reduce its incidence [20,33]. On the other hand, targeted interventions involving psychotherapeutic sessions with adolescents are of interest, as they directly contribute to the improvement of personality traits and overall development. However, the actions must be directed at the family, the teachers, and the rest of the school community. It is therefore necessary to articulate training actions on the responsible and safe use of the internet, the collaboration of the family with the school in educational activities, and, above all, the importance of promoting safe environments in schools [20,34,35].

The limitations of this study include the fact that it was a cross-sectional study focused on one age group, with educational centers selected for convenience, as well as the fact that it focused on the use of questionnaires, which may affect social desirability. It would be of interest to extend the study to include other influential variables that would help to increase the variance explained. Furthermore, conducting longitudinal studies would provide a more precise profile.

## 5. Conclusions

Cyberbullying has been identified as a significant mental health threat by researchers. Adolescents who assume direct roles in cyberbullying exhibit psychopathological personality traits or behaviors throughout their high school years, irrespective of gender.

The conclusions derived from this study offer insight into the phenomenon of cyberbullying and its impact on psychopathological behaviors. This knowledge will enable school administrators, teachers, family members, and other educational stakeholders to collaborate and devise a comprehensive framework to address and prevent cyberbullying across all forms.

It is considered that this work will support teachers in recognizing and responding appropriately to cyberbullying, thereby enhancing the ability of educational and social institutions to tackle issues associated with cyberbullying, discrimination, and violence. Additionally, preventative approaches should be examined from a interdisciplinary standpoint, ultimately creating an comprehensive support structure for students, not only focusing on victims but also on aggressors and bystanders.

## Figures and Tables

**Table 1 healthcare-11-03162-t001:** Descriptive data of means, standard deviation, minimum, and maximum of variables.

	N	Min.	Max.	M	SD
CB as victim	280	15.00	37.00	16.38	2.85
CB as aggressor	280	15.00	44.00	15.62	2.41
CB as aggressive victimization	280	30.00	81.00	32.00	4.90
CB as bystander	280	15.00	43.00	17.00	3.54
Hypochondriasis	280	0.00	20.00	6.70	4.97
Suicidal depression	280	0.00	24.00	7.04	5.36
Agitation	280	5.00	21.00	11.68	2.85
Anxious depression	280	3.00	19.00	10.91	3.09
Low-energy Depression	280	0.00	23.00	10.03	5.04
Guilt–resentment	280	0.00	23.00	10.23	5.10
Apathy–withdrawal	280	0.00	20.00	7.75	3.99
Paranoia	280	1.00	20.00	9.07	3.84
Psychopathic deviance	280	6.00	20.00	13.18	2.84
Schizophrenia	280	0.00	19.00	6.26	4.41
Psychasthenia	280	5.00	22.00	11.82	3.05
Psychological maladjustment	280	0.00	21.00	7.97	4.64

Note: CB: cyberbullying. Means (M), standard deviations (SD).

**Table 2 healthcare-11-03162-t002:** Bivariate correlation between cyberbullying variables and psychopathological behaviors.

	CB as Victim	CB as Aggressor	CB as Aggressive Victimization	CB as Bystander
Hypochondriasis	0.386 ***	0.179 **	0.314 ***	0.203 **
Suicidal depression	0.348 ***	0.177 **	0.291 ***	0.164 **
Agitation	0.027	0.080	0.054	0.156 **
Anxious depression	0.216 ***	0.062	0.157 **	0.014
Low-energy Depression	0.294 ***	0.128 **	0.235 ***	0.159 **
Guilt–resentment	0.349 ***	0.095	0.252 ***	0.168 **
Apathy–withdrawal	0.261 ***	0.157 **	0.229 ***	0.115 *
Paranoia	0.446 ***	0.190 ***	0.354 ***	0.270 ***
Psychopathic deviance	0.270 **	0.073	0.032	0.195 ***
Schizophrenia	0.445 ***	0.235 **	0.376 ***	0.249 ***
Psychasthenia	0.278 ***	0.144 **	0.234 ***	0.244 ***
Psychological maladjustment	0.341 ***	0.152 **	0.274 ***	0.159 **

* *p* < 0.05; ** *p* < 0.01; *** *p* < 0.01.

**Table 3 healthcare-11-03162-t003:** Hierarchical regression analysis for predicting the role of victim in cyberbullying.

Model	R²	Adjusted r-Squared	R² Change	F Change	F
1	0.011	0.003	0.011	1.373	1.373
2	0.285	0.254	0.274	11.128 ***	9.269 ***

Note: *** *p* < 0.001.

**Table 4 healthcare-11-03162-t004:** Hierarchical regression analysis for the prediction of the role of aggressor in cyberbullying.

Model	R²	Adjusted r-Squared	R² Change	F Change	F
1	0.004	−0.005	0.004	0.428	0.428
2	0.091	0.052	0.088	2.809 **	2.338 **

Note: ** *p* < 0.01.

**Table 5 healthcare-11-03162-t005:** Hierarchical regression analysis for the prediction of the bystander role in cyberbullying.

Model	R²	Adjusted r-Squared	R² Change	F Change	F
1	0.010	0.001	0.010	1.163	1.163
2	0.146	0.109	0.136	4.635 ***	3.969 ***

Note: *** *p* < 0.001.

**Table 6 healthcare-11-03162-t006:** Hierarchical regression analysis for the prediction of the role of aggressive victim in cyberbullying.

Model	R²	Adjusted r-Squared	R² Change	F Change	F
1	0.005	−0.004	0.005	0.545	0.545
2	0.196	0.162	0.192	6.941 ***	5.683 ***

Note: *** *p* < 0.001.

## Data Availability

The data presented in this study are available upon request to the corresponding author. The data are not publicly available due to the privacy of the participants. The language of the data is Spanish.

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
