# Peer review of "Cyberbullying and Psychopathological Behaviors in Spanish Secondary Education Students"

_healthcare, 2023, doi:10.3390/healthcare11243162_

Round 1

Reviewer 1 Report

Comments and Suggestions for Authors

Author Response

Dear reviewer: First of all, we would like to thank you for the comments, suggestions and modifications proposed by the Scientific Council of the journal regarding our article. We have carried out a review of the text to which we have added the modifications in yellow and this document clarifies some aspects of the text. See more detail in the attached letter.

Reviewer 2 Report

Comments and Suggestions for Authors

Thanks for giving me the opportunity to read this paper.

The paper presents a research carried out with 280 students looking for the association between roles involved in cyberbullying and psychopathological behaviours. The exploration of these associations is a very important topic and with high interest within the field of educational psychology.

The paper is very well structured. The introduction is well presented, the most relevant authors in the field are referred, and the need to research cyberbullying, roles involved, and psychopathological behaviours associated in the four roles studied (victim, aggressor, bystanders, and aggressive victim) is well stated.

The method is well presented especially regarding the instruments (items, alphas); nevertheless, the alpha of the subscales of psychopathological behaviours questionnaire should be presented, as the results are presented per each subscale. The ethical procedure with all the permissions required is very well presented too, but more information about how many families declined to participate could provide a very interesting view.

The sample is not well defined; cyberbullying is related to the social group but in the paper the information about how many schools participated, how many classes, the number of the groups, the number of students per class, if the whole class was participating, or only some students randomly selected... are questions that should be responded.  It would be nice if the authors can provide more information about the participants.

The presentation of the results should be improved. For example, a descriptive table showing the means, standard deviation, minimum, maximum per each variable would be very valuable to understand better the results.

Regarding the results, the R2 is extremely low in all the roles except for the aggressors, which is very interesting information that the authors do not explain enough; and because the Table with the descriptive is missing, it is not possible to argue that maybe this results is because the most affected role in bullying are the aggressors, or because the other roles maybe were not enough clarify or defined.

A relevant limitation in order to generalise the results is related to the way participants were selected to take part, if all of them, or they were randomly selected, and that limitation is not explained either.

Therefore, more data should be presented, to understand the results, and have a deeper and more complete discussion and conclusion.

Author Response

(The authors gave the same response as above.)

Reviewer 3 Report

Comments and Suggestions for Authors

Dear author, congratulations for the present manuscript that explores an important issue.

However before the manuscript it is ready for publication I have several recommendations that are in the main document in attached. By biggest concern it is the fact that the authors explored predictors but the theoretical support are the consequences, namely psychopathological consequences.

the discussion it is also poor without justifying the results.

Comments on the Quality of English Language

Minor editing of English language required

Author Response

(The authors gave the same response as above.)

Reviewer 4 Report

Comments and Suggestions for Authors

Title:

Cyberbullying, and Psychopathological Behaviors in Compulsory Secondary Education Students

The reviewer’s comments

The reviewer would like to see some revisions made to your manuscript.

1)     Increase the discussion of literature on cyberbullying, and psychopathological behaviors in compulsory secondary education students

2)     Data analysis is too simple, please explain it in detail.

3)     Discussion and literature review link, and generate meaning.

4)     Please strengthen the conclusion and implications. Good finding suggestions for future practitioners and researchers.

5)     Major revision.

Comments on the Quality of English Language

Title:

Cyberbullying, and Psychopathological Behaviors in Compulsory Secondary Education Students

The reviewer’s comments

The reviewer would like to see some revisions made to your manuscript.

1)     Increase the discussion of literature on cyberbullying, and psychopathological behaviors in compulsory secondary education students

2)     Data analysis is too simple, please explain it in detail.

3)     Discussion and literature review link, and generate meaning.

4)     Please strengthen the conclusion and implications. Good finding suggestions for future practitioners and researchers.

5)     Major revision.

Author Response

(The authors gave the same response as above.)

Round 2

Reviewer 1 Report

Comments and Suggestions for Authors

I am happy to see the improvement of this manuscript. It displays interesting data, better framed in the context of the study. Thus, I now recommend the Editor(s) consider this paper for publication in the prestigious journal "Healthcare". However, I have some more suggestions for the authors:

1. Even if I am not a native speaker, I think that language and punctuation must be improved. Thus, I recommend professional and academic editing. For example, line 197 should be: The design was non-experimental, quantitative, transactional, and descriptive.

2. The authors should read the paper carefully again and correct any kind of typos. For example, in Table 5, column “Model” the number should be 1 and 2, I guess, and not 1 and 3

I wish the authors the best.

Author Response

Dear reviewer:

First of all, we would like to thank you for the comments regarding our article. Following the requested suggestion we have revised with a native colleague the wording of the paper. However, we will be happy to send the manuscript to grammar review at the publisher itself for further revision when the manuscript does not need further changes.

Following the suggestion given by the reviewer, we have thoroughly revised the manuscript. We appreciate that he noticed the error in Table 5, which has been corrected in the updated version.

Hopefully it will be in accordance with the reviewer's request. Any comments are welcome. The new changes have been added in green color in the document.

Best regards,

Reviewer 2 Report

Comments and Suggestions for Authors

Dear author

It would be convenient to include the table with the correlations between variables-

Including that the paper is fine to be published

Thanks

Author Response

Dear reviewer:

First of all, we would like to thank you for the comments regarding our article. Following the requested suggestion we have incorporated a correlation table. Hopefully it will be in accordance with the reviewer's request.

We have thoroughly revised the English grammar of our manuscript.

Any comments are welcome. The new changes have been added in green color in the document.

Best regards,

Reviewer 3 Report

Comments and Suggestions for Authors

Dear authors, thank you for you efforts, but the results aren't clearly presented. There are some B that are negative and the authors don't explains in the discussion.

Author Response

Dear reviewer:

First of all, we would like to thank you for the comments on our article.

We reviewed the data obtained and the reviewer was right since the negative sign was an error when transferring the data to the text. We regret that we did not clarify this in the previous review letter. We thank the reviewer for noticing this error and correcting it.

We have thoroughly revised the English grammar of our manuscript. The new changes have been added in green in the document.

Any comment is welcomed.

Sincerely,

Reviewer 4 Report

Comments and Suggestions for Authors

Title:

 Cyberbullying, and psychopathological behaviors in Compulsory
Secondary Education students

The reviewer’s comments

Thanks to the author for the correction.Revisions or explanations are all made according to the suggestions of the reviewers. Accept in present form.

Comments on the Quality of English Language

Title:

 Cyberbullying, and psychopathological behaviors in Compulsory
Secondary Education students

The reviewer’s comments

Thanks to the author for the correction.Revisions or explanations are all made according to the suggestions of the reviewers. Accept in present form.

Author Response

Dear reviewer:

First of all, we would like to thank you for the comments regarding our article. We are glad to hear that the article has been improved in the current version.

We have thoroughly revised the English grammar of our manuscript. The new changes have been added in green in the document.

Sincerely,

Best regards,